

# Multi-classification of disease induced in plant leaf using chronological Flamingo search optimization with transfer learning

Malathi Chilakalapudi and Sheela Jayachandran

SCOPE, VIT-AP University, Amaravathi, Andhra Pradesh, India

## ABSTRACT

Agriculture is imperative research in visual detection through computers. Here, the disease in plants can distress the quality and cultivation of farming. Earlier detection of disease lessens economic losses and provides better crop yield. Detection of disease from crops manually is an expensive and time-consuming task. A new scheme is devised for accomplishing multi-classification of disease using plant leaf images considering the chronological Flamingo search algorithm (CFSA) with transfer learning (TL). The leaf image undergoes pre-processing using Adaptive Anisotropic diffusion to discard noise. Here, the segmentation of plant leaf is done with U-Net++, and trained by the Moving Gorilla Remora algorithm (MGRA). The image augmentation is further applied considering two techniques namely position augmentation and color augmentation to reduce data dimensionality. Thereafter the feature mining is done to produce crucial features. Next, the classification in terms of the first level is considered for classifying plant type and classification in terms of the second level is done to categorize disease using convolutional neural network (CNN)-based TL with LeNet and it undergoes training using CFSA. The CFSA-TL-based CNN with LeNet provided better accuracy of 95.7%, sensitivity of 96.5% and specificity of 94.7%. Thus, this model is better for earlier plant leaf disease detection.

## INTRODUCTION

India represents a country that has fame in the domain of agriculture in which the majority of the population relies on agriculture. The research in the domain of agriculture is aimed at elevating productivity and quality of food at less cost and more profit. The agriculture production model is a result of complicated interactions of soil, agrochemicals, and seeds. Fruits and vegetables are the imperative products of agriculture. The disease indicates impairment to normal plant state which enhances or interrupts its crucial operations like transpiration, photosynthesis, fertilization, pollination, and germination. Hence, the earlier treatment of disease in plants is termed a crucial task. The farmer needs repeated monitoring from expertise which can be costly and take more time. Hence, determining a quick, cost-effective, and precise technique for automatically determining the disease through the symptoms of the plant leaf is of huge importance (*Gavhale & Gawande, 2014*).

Corresponding author
Sheela Jayachandran,
sheela.j@vitap.ac.in

The disease of plant leaves are imperative cause of loss production and the detection of disease in plant leaves is also a complex process in the agriculture domain (*Annabel, Annapoorani & Deepalakshmi, 2019*). For diagnosing the batches of leaf, there exists some bewilderment because of the similarities amidst the size of batches, and color which only specialists can detect. The initial step in fighting alongside leaf batches is sufficient detection of its presence which is a precise diagnosis. The abnormal symptoms are the suggestion to the existence of a disease and thus can be considered as an aid in treatment (*Jagtap & Hambarde, 2014*; *Geetha et al., 2020*). The disease in plants can cause the serious eruption of diseases that lead to huge-scale death and food shortages. It is evaluated that the eruption of helminthosporiose of rice led to huge food loss and the deaths of several people. The impact of plant disease tends to be devastating and some of the cultivation of crops is ditched. Observation from the naked eye through expertise is a major technique employed in practice for detecting and identifying disease in plants, but it needs repeated expert monitoring that can be costly while dealing with huge farms (*Arivazhagan et al., 2013*).

Contemporary unrefined farming is acquiring a reputation in the domain of agriculture considering several developing nations. There exist several issues that arise in farming because of several environmental aspects and this disease in plant leaves is termed to be a strong aspect that causes deficiency in the quality of agricultural products. The aim is to alleviate this problem with machine learning (ML) models (*Bayram, Bingol & Alatas, 2022*; *Bingöl, 2022a*, *2022b*). Several ML and segmentation models are devised for the categorization and discovery of diseases in plants through leaf images (*Subramanian et al., 2022*; *Krishnamoorthy et al., 2021*; *Sathishkumar et al., 2020*). These methods have built a way to discard the issues but the issues being confronted are the performance outcomes generated (*Hossain, Hossain & Rahaman, 2019*). Many methods are devised for detecting and classifying the plant leaf disease among which the k-nearest neighbor classifier tends to be an effective one (*Jasim & Al-Tuwaijari, 2020*). The studies revealed that deep learning models are effectual techniques for classifying diseases in plants. An automatic model aimed at aiding plant disease treatment considering the existence and noticeable symbols of plants can help the learners in growing tasks and also help the professionals diagnose diseases (*Sladojevic et al., 2016*; *Grinblat et al., 2016*). There exist several types of research which are performed each year in cultivating the crop using computer vision and image processing (*Al-Tuwaijari, Mohammed & Rahem, 2018*; *Jasim & Al-Tuwaijari, 2020*; *Sachdeva, Singh & Kaur, 2021*).

## MOTIVATION

Organic farming has become more general in several countries that follow agricultural practices. There exists a huge number of problems that happen in plant growth because of several environmental aspects. The disease in crops can cause a reduction in productivity and thus detection of crop disease in the starting stage can offer huge benefits in the domain of agriculture. The challenges in the existing methods are:

- The segmentation may become a complex process because of contrast, scale, and shape alterations.
- The images of low contrast impact the detection performance elevate the computational cost and minimize the classification accuracy.
- The manual elucidation needs a huge quantity of work and expertise in detecting the disease and also needs a huge time for processing.

These challenges are considered the motivation for developing a new model for classifying plant disease using the leaf images.

The aim is to design multi-classification with leaf images using the chronological Flamingo search algorithm (CFSA) with convolutional neural network (CNN) based transfer learning (TL) and LeNet. The article's chief contribution is:

- **Designed CFSA-TL-based CNN with LeNet for first-level classification:** The classification in terms of the first level includes the classification of plant leaf type using CFSA-TL-based CNN with LeNet. Here, the TL-based CNN with LeNet is trained with CFSA, which is developed by unifying the chronological concept in the Flamingo search algorithm (FSA).
- **Developed CFSA-TL-based CNN with LeNet for second-level classification:** The classification in terms of the second level includes the classification of plant leaf disease using CFSA-TL-based CNN with LeNet. Here, the TL-based CNN with LeNet is trained with CFSA and is developed by unifying the chronological concept in FSA.

The remaining sections are arranged as follows: "Motivation" defines previously developed plant leaf disease categorization models. "Proposed CFSA-Based TL for Multi-Classification of Plant Leaf Disease" illustrates the proposed model to classify the plant leaf disease. "Discussion of Outcomes" provides the analysis of outcomes in revealing the efficacy of each method and finally, "Conclusion" concludes with TL-based CNN with LeNet.

## Literature survey

*Hussain et al. (2022)* designed a model for the detection of diseases from cucumber leaves. The model was developed based on deep learning (DL) and includes fusion and collection of optimum features. Here, a visual geometry group (VGG) was used. The feature obtained was fused with a maximum fusion scheme and optimum features were selected with the Whale Optimization algorithm and classified with supervised learning. However, this technique was imperfect in handling other databases. *Jadhav, Udup & Patil (2019)* developed a model using k-nearest neighbors (KNN) classifiers, and a multiclass support vector machine for the detection and categorization of soybean diseases with color images. The thresholding was applied for extracting interesting regions. The Incremental K-means clustering was adapted for segmentation and finally, support vector machine (SVM) and KNN were utilized to classify disease. The method suffered from overfitting problems. *Singh & Kaur (2021)* devised a technique for detecting and classifying the disease that occurred in potato plants. Here, the consistent data set was adapted which was known as

the Plant Village database. Here, the K-means scheme was considered for segmenting images, and the gray-level co-occurrence matrix concept with multi-class SVM was applied for classification. This method provided poor accuracy while dealing with huge data. *Islam et al. (2017)* devised a technique that unified machine learning and image processing for permitting disease diagnosis through leaf images. The automatic technique classified the disease of the potato plant. The SVM was utilized for classifying disease and provided better accuracy. It was not appropriate for handling other datasets. *Tiwari, Joshi & Dutta (2021)*, designed a deep learning (DL)-based method for identifying and classifying the disease in plants with leaf images acquired through different resolutions. Here, the dense convolutional neural network (dense CNN) was trained using huge plant leaf images considering data from different countries. The method was not able to expand the plant leaf database for handling complex platforms. *Roy & Bhaduri (2021)* devised a deep learning-enabled object detection technique for classifying plant disease. The method helps to provide accurate discovery and fine-grained detection of disease. Moreover, the model was enhanced to optimize both speed and accuracy. However, this method was not suitable for handling real platforms. *Atila et al. (2021)*, devised EfficientNet for classifying disease from plant leaves. Here, the PlantVillage dataset was utilized for training the models. Each model was trained with a different set of images. Here, the concept of transfer learning was used where all layers were trained for performing classification. However, this method failed to extend the disease database by elevating the diversity of plants considering different sets of classes. *Lakshmi & Savarimuthu (2021)* devised a deep learning framework for automatic plant disease detection and segmentation (DPD-DS) considering an enhanced pixel-wise mask-region-based convolution neural network (CNN). It utilized a region convolution neural network (R-CNN) to save memory and cost. It helps to elevate detection accuracy. However, it did not adopt an ensemble network for detecting disease in plant leaves. *Bayram, Bingol & Alatas (2022)* established an artificial intelligence technique for automatically detecting tomato leaf disease. Here, for the classification Inceptionv3, Resnet50, Efficientb0, Shufflenet, Googlenet, and Alexnet models were used. Then, the feature maps were gathered from the tomato images. The neighborhood component analysis (NCA) was used in the feature extraction. However, a single disease dataset was used in this research.

# PROPOSED CFSA-BASED TL FOR MULTI-CLASSIFICATION OF PLANT LEAF DISEASE

Figure 1 exposes the overlook of the multi-classification of plant leaf disease framework considering CFSA. This article provides a TL-based CNN with LeNet considering CFSA for multi-classification of plant leaf disease using images. Firstly, the leaf image is acquired and provided for pre-processing to eliminate noise using Adaptive Anisotropic diffusion. Then the segmentation of plant leaf is implemented using the U-Net++, which is trained with RGWMA and is produced by the combination of GTO, EWMA, and ROA. Then, the image augmentation is done which is classified into position augmentation and color augmentation. Then, feature extraction is done with the hybrid opponent color local binary pattern (OCLBP) based discrete wavelet transform (DWT) using the histogram of

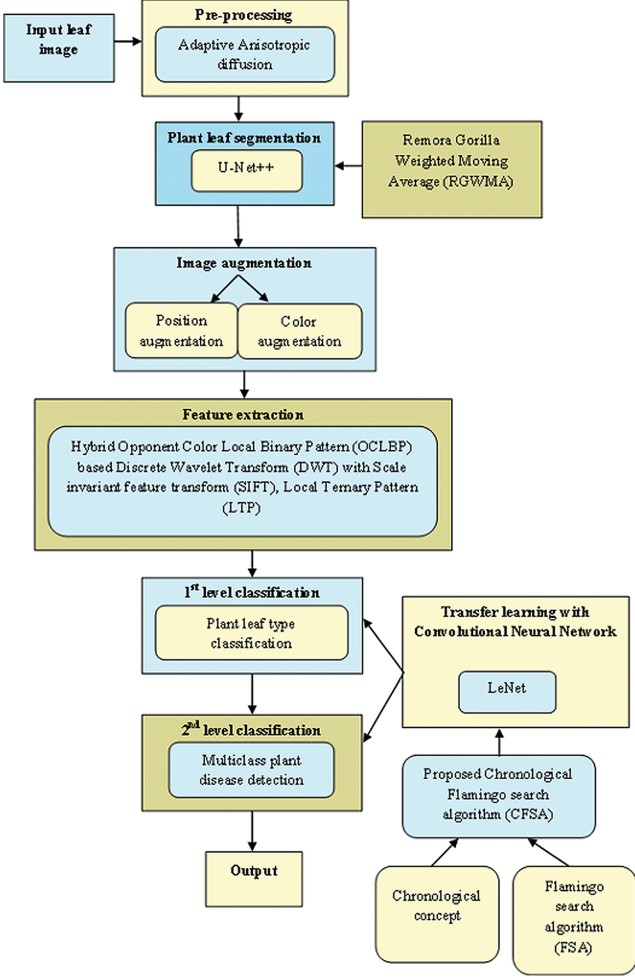

**Figure 1 Overlook of multi-classification of plant leaf disease model with CFSA.**

oriented gradients (HOG), scale invariant feature transform (SIFT) and local ternary pattern (LTP). Next, the first level classification is considered as plant type classification and the second level classification is considered as the multi-class plant disease detection, both classifications are done by using a transfer learning-based CNN with LeNet. For the first and second classifications, the training process is done with CFSA. The CFSA produced is the blending of chronological concepts in FSA and finally, the output is noted.

## Image acquisition

Research for the automatic discovery of diseases in plants has been of great interest among researchers for several years. A model is developed for detecting several diseases in plants considering plant leaf images. Due to increasing plant images, less expertise and knowledge cannot fulfill the requirements of huge-scale image processing. Thus, an automated detection of plant leaf has gained more focus. Due to the design of imaging techniques, people can simply attain clear images of plants and the computer-assisted detection of plant images is a major hotspot. Imagine a database $F$ having a set of plant leaf images $e$ and is represented as

$$F = \{\chi_1, \chi_2, \cdots, \chi_v, \cdots, \chi_e\} \tag{1}$$

where, $e$ depicts total images and $\chi_v$ symbolizes $v^{th}$ image.

## Pre-processing with adaptive anisotropic diffusion

In the pre-processing phase, the inputted plant leaf image $\chi_v$ is considered. The pre-processing is adapted on the provided image in a sequence which makes it appropriate for better processing. The fundamental pre-processing phase is to resize the provided image. The initial image size is huge which makes it complex and takes more processing time. Hence, the application of pre-processing is done to eliminate noise and make it apt for improved processing. Thus, the adaptive anisotropic diffusion (*Tang et al., 2007*) is adapted for pre-processing. It is an effective smoothing procedure. Also, it is mainly used for noise removal and edge-preserving. The data loss and image blur problems are avoided in this model. Here, the evaluation of the edge map and the selection of k in diffusion coefficients are more imperative. The imperative selection of these two things will generate an improved anisotropic diffusion. If k is high, then the edge preservation will be best, but noise will not be discarded. If k is small, then noise will be discarded, but edges will be blurred. Hence, how to devise k is an imperative parameter in the diffusion technique. A suitable k is to divide the noise through the edges in an effective manner. The operator of gradient magnitude is sensitive to noise particularly whenever the edge strength is weak. Thus, the gradient magnitude is utilized as an edge map and is stated as

$$k(q, r) = \sqrt{\frac{|\nabla B_f(q, r)|^2 + |\nabla B_D(q, r)|^2 + |\nabla B_K(q, r)|^2 + |\nabla B_M(q, r)|^2}{4}} \tag{2}$$

$$\nabla B_D(q, r) = B(q, r+1) - B(q, r) \tag{3}$$
$$\nabla B_f(q, r) = B(q, r-1) - B(q, r) \tag{4}$$
$$\nabla B_K(q, r) = B(q+1, r) - B(q, r) \tag{5}$$
$$\nabla B_M(q, r) = B(q-1, r) - B(q, r). \tag{6}$$

The evaluation of diffusion coefficients is explored as

$$d(k) = \exp\left(-\left[\frac{k(q, r)}{m}\right]^2\right) \tag{7}$$

$$d(k) = \frac{1}{1 + \left[k(q, r)^2 - m^2\right]/\left[m^2(1 + m^2)\right]}. \tag{8}$$

Assume the distribution of noise in an image and consider a homogenous region with noise and can consider that $\nabla B_D(q, r)$, $\nabla B_f(q, r)$, $\nabla B_K(q, r)$, and $\nabla B_M(q, r)$ poses the same zero mean normal distribution like $\aleph(0, \sigma^2)$ and hence equation becomes,

$$\frac{4}{\sigma^2} * k(q, r)^2 = \frac{1}{\sigma^2}\left(|\nabla B_f(q, r)|^2 + |\nabla B_D(q, r)|^2 + |\nabla B_K(q, r)|^2 + |\nabla B_M(q, r)|^2\right) \tag{9}$$

It poses a chi-squared distribution having four freedom degrees.

Once the noise gradient distribution is obtained, compute the choice of $m$. Consider that $q, r$ indicates the pixel coordinates of the image and one can utilize Eq. (9) to generate the gradient map. If the point belongs to the edge, then the gradient map becomes an outlier in four freedom degrees. Hence the threshold $m$ is chosen for rejecting those outliers and $m$ is generated as

$$\rho\left(k(q, r)^2 > m^2\right) \leq \eta. \tag{10}$$

The aforesaid expression is rewritten as

$$\rho\left(\frac{4}{\sigma^2} * k(q, r)^2 > m^2 * \frac{4}{\sigma^2}\right) \leq \eta \tag{11}$$

where, $\eta$ stands for significance level. Hence, the $m$ is set as

$$m = d\sigma \tag{13}$$

where, $\sigma^2$ represents the variance of noise gradient and $d$ denotes constant. Hence, the pre-processed outcome generated through adaptive anisotropic diffusion is notified as T.

## Segmentation of plant leaf with Moving Gorilla Remora algorithm (MGRA)-based U-Net++

Thus, the pre-processed outcome T is provided as segmentation input. The segmentation function aimed to mine the complete leaf region over the background. For improving the reliability and accuracy of mining, it is essential for the model to be in capacity for depicting the features contained in the image that are fine-grained and alteration in size and shape. The U-Net++ (*Fan et al., 2022*) is adapted for performing the segmentation of plant leaves with MGRA-based U-Net++. The structure of U-Net++ and its training module is stated in subsections.

### *Overview of U-Net++*

A U-Net++ (*Fan et al., 2022*) is termed as a backbone platform for accomplishing the segmentation. The U-Net++ is developed using U-Net and is devised for meeting the needs of precisely segmenting the images. This model substitutes the plain skip connection in the place of nested and dense skip connections and it obtains fine data. It poses the ability to find leaves having various sizes considering the feature maps having various scales. For handling features of leaves, the U-Net++ is suitable for attaining segmentation of plants. The U-Net++ comprises three major modules, namely encoding, decoding, and dense concatenation of cross-layers.

The outputted features through the encoder are combined with the upcoming encoder layer through the features of up-sampling amid layers. The fused outputs are combined with equivalent up-sampled features of consequent layers and are repeated till there is no equivalent module in the upcoming layer. The unified feature maps are described by

$$b^{s,t} = \begin{cases} \Im(b^{s-1,t}) & t = 0 \\ \Im\left(\left[[b^{s,p}]_{p=0}^{t-1}, \eta(b^{s+1,t-1})\right]\right) & t > 0 \end{cases} \tag{14}$$

where $\Im(\cdot)$ delineates convolution function, $\eta(\cdot)$ states the upsampling layer, $[\,]$ is the concatenation layer. Node residing at level $t = 0$ receives input through prior encoder layer while nodes at level $t = 1$ receives the encoder and input of sub-network from two successive levels and nodes $t > 1$ receives $t + 1$ of which $t$ input are termed as outputs of prior $t$ nodes in similar skip pathways and the final input is up-sampled outcome from low skip pathway. The dense skip connections amid layers of similar size pass the outcome of present modules to all equivalent modules and combine it using other inputted features. Hence complete U-Net++ fusion model is modelled in the format of an inverted pyramid in which the intermediate layer comprises more precise localization data whereas the in-depth layer acquires pixel-level class data. The purpose is to segment the plant image into binary by labeling it as background and foreground as 0 and 1.

### U-Net++ training with MGRA

The U-Net++ training is devised through MGRA by updating the weights of U-Net++. The update expression of the remora optimization algorithm (ROA) host feeding module (*Jia, Peng & Lang, 2021*) is induced with the Gorilla Weighted Moving Average algorithm (GWMA) to design MGRA. Here, GWMA is developed by combining GTO (*Xiao et al., 2022*) and EWMA (*Saccucci, Amin & Lucas, 1992*). Hence, the MGRA's update expression is presented as,

$$
\begin{aligned}
G(j+1) = \Bigg[ & \frac{1}{\mu} \big[ \mu \times (\rho_3 - L) \times G_N(j) + P \times Q \times G^\varepsilon(j) - (1-\mu) \times G^\varepsilon(j-1) \big] \\
& - \frac{O\beta G_{best}}{(1+O)} + G(j) \Bigg] \times \left( \frac{1+O}{2+O} \right).
\end{aligned}
\tag{15}
$$

Thus, $G_{best}$ stands for the best solution, $\beta$ delineates the remora factor, and $j$ expresses current iteration, $G(j)$ indicates the current position vector of each gorilla, $\rho_3$, implies arbitrary number amidst 0 and 1, $L$ articulate constant, $G_N(j)$ states arbitrarily selected gorilla sites in the current population, $Q$ signifies row vector in problem computation using the rate of the unit are randomly built-in $[-L, L]$, $\mu$ expresses smoothing factor, $G^\varepsilon(j)$ signifies evaluated location of search agent at $j^{th}$ iteration, $G^\varepsilon(j-1)$ notify estimated search agent position at $(j-1)^{th}$ iteration, $P$ and $O$ delineate constants. Hence, the segmented outcome attained with RGWMA-based U-Net++ is explicated as $E$.

## Augmentation of image

The segmented outcome $E$ is provided to this phase. It is useful when one is provided a database with very less instances. Moreover, this process aids in fighting overfitting and enhances the efficiency of deep networks for dealing with several tasks. It aids in making data rich and adequate hence making the model perform better and precisely. It helps to minimize operational costs considering various kinds of transformations. Here, position and color augmentation methods are applied and are examined below:

### Position augmentation

These methods affected the location of pixel values to build augmented images. The amalgamation of four transformations, like padding, rotation, translation, and affine transformation are utilized in this research.

**a) Padding**

Padding is the process of adding a border around the image. It helps to build space over the element's content inside any described borders. It indicates the blank space over the image. This augmented outcome is notified by $Z_1$.

**b) Rotation**

Rotation is a process of rotating an image with certain angles and orientations. It utilizes two major attributes rotation angle and the point through which the rotation is performed. It is essential for extracting features and matching features. This augmented result is notated by $Z_2$.

**c) Translation**

Translation is the process of transforming an image from one domain to another domain in which the aim is to learn the mapping amid input and output images. This method aims to learn the mapping relationship amid the input and target image for suitably transforming the former to the latter. This augmented output is signified by $Z_3$.

**d) Affine transformation**

Affine transformation indicates a linear mapping technique that preserves points, planes and direct lines. The group of parallel lines remained parallel after performing an affine transformation. It is used to correct geometric distortions and deformations which happen with non-ideal angles of the camera. This outcome is expressed by $Z_4$.

### Color augmentation

This method alters the properties of color considering original images to build augmented mages. Here, the unification of four properties is utilized for producing novel images like contrast, hue, and saturation.

**a) Contrast**

Contrast refers to a term that depicts the change in contrast amid light and dark colors. It indicates the quantity of color or differentiation between gray scales that exist amid several image features. The image with a high level of contrast exposes a huge degree of color variation compared to those with low contrast. The contrast output is signified by $Z_5$.

**b) Hue**

The hue refers to a wavelength in the visible light spectrum in which the output of energy from the source tends to be greatest. This augmented outcome is notated as $Z_6$.

**c) Saturation**

Saturation indicates an expression to depict the relative bandwidth of the visible outcome through the light sources. This augmented result is notified as $Z_7$.

Hence, the augmentation vector $Z$ produced is modeled as

$$Z = \{Z_1, Z_2, \cdots, Z_7\}. \tag{16}$$

## Obtain crucial feature

Here, the augmented image vector $Z$ is given as input. Feature extraction indicates an imperative step in constructing the pattern categorization and its goal is to extract the pertinent data which helps to characterize each class. Here, the pertinent features are obtained through the images to form a feature vector. These feature vectors are further used by the classifier for recognizing the target. It is easy for the classifier to classify different classes by adapting the features as it permits easy to distinguish. Here, features like OCLBP-based DWT, SIFT, and LTP features are mined:

### a) OCLBP-based DWT

The augmented image is fed to DWT and it is divided into LL, LH, HL, and HH bands wherein the HH band contains noise and thus it is prevented. Hence LL, LH, and HL bands are adapted with OCLBP (*Vishnoi, Kumar & Kumar, 2022*) and finally concatenated and applied with HoG to establish a feature vector.

OCLBP (*Vishnoi, Kumar & Kumar, 2022*) indicates a joint color-texture feature that helps to compare gray scale and color textures. Here, pairs of colors like red-green and yellow-blue are acquired by humans and called opponent colors. These opponent colors are determined by adapting LBP on center and adjacent pixels with opposite color channels. The texture $\tau$ is adapted as a distribution $\tau \approx \gamma(\alpha(v_\kappa - v_\delta), \cdots, \alpha(v_{\kappa-1} - v_{\delta-1}))$. The local texture considering the image around $\Re(\vartheta_\delta, \hbar_\delta)$ is described as

$$LBP_{\zeta,\Re(\vartheta_\delta,\hbar_\delta)} = \sum_{\kappa=0}^{\kappa-1} \alpha(v_\kappa - v_\delta)2^\kappa \tag{17}$$

$$\alpha(j) = \begin{cases} 1, & \vartheta \geq 0 \\ 0, & \vartheta \leq 0. \end{cases} \tag{18}$$

Hence, the texture of the image is described approximately as

$$\tau = \gamma\left(LBP_{\zeta,\Re(\vartheta_\delta,\hbar_\delta)}\right). \tag{19}$$

HoG (*Vishnoi, Kumar & Kumar, 2022*) offers information regarding the occurrence of orientation-related gradients in RoI or local regions. The evaluation of gradient $R$ and direction $\phi$ is modeled in a generalized way as

$$|R| = \left(R_\vartheta^2 + R_\hbar^2\right)^{\frac{1}{2}} \tag{20}$$

$$\phi = \tan^{-1}\frac{R_\vartheta}{R_\hbar} \tag{21}$$

where $R_\vartheta$ and $R_\hbar$ represent gradient along $\vartheta$ and $\hbar$ directions. The image is split into various square cells or areas with specific sizes. This feature is explained as $V_1$.

### b) SIFT

SIFT (*Vishnoi, Kumar & Kumar, 2022*) offers local key features considering the objects that are unchanged against the transformations of scale. It involves three steps namely determining the scale-space maxima and keypoints, orientation assignment, and key point descriptor.

*Determination of scale-space maxima and key points:* The operation utilized to find key points is termed scale-space and is expressed as $A(a, c, \sigma)$. It is termed as a convolution of Gaussian kernel $C(a, c, \sigma)$ and image $H(a, c)$ such that

$$A(a, c, \sigma) = C(a, c, \sigma) * H(a, c) \tag{22}$$

where, $*$ denotes convolution amid $a$ and $c$, and $\sigma$ stands for the standard deviation of various scales such that

$$C(a, c, \sigma) = \frac{1}{2\pi\sigma^2} \exp^{-\frac{a^2+c^2}{2\sigma^2}}. \tag{23}$$

The difference of Gaussians (DOG) $S(a, c, \sigma)$ is evaluated as

$$S(a, c, \sigma) = A(a, c, T\sigma) - A(a, c, \sigma) \tag{24}$$

where $T$ is constant which divides two successive smooth images.

*Orientation assignment:* Hence, the gradient magnitude $V(a, c)$ and orientation $W(a, c)$ considering smoothed image at scale $\sigma$ is given by

$$V(a, c) = \left\{ (A(a+1, c) - A(a-1, c))^2 + (A(a, c+1) - A(a, c-1))^2 \right\}^{\frac{1}{2}} \tag{25}$$

$$W(a, c) = \tan^{-1} \frac{A(a, c+1) - A(a, c-1)}{A(a+1, c) - A(a-1, c)}. \tag{26}$$

*Key point descriptor:* It is established for every key that poses a stable orientation, scale, and position. This SIFT feature is signified by $V_2$.

*c) LTP*

LTP (*Vishnoi, Kumar & Kumar, 2022*) is an expansion of LBP in which the center pixel and its adjacent pixels are done in three unique zones. Considering three zones, the LTP histogram is generated. The LTP operator is stated as

$$LTP_{\zeta, \Re}(s, t) = \sum_{\kappa=0}^{\kappa-1} \alpha(v_\kappa - v_\delta) 3^\kappa \tag{27}$$

where

$$\alpha(j) = \begin{cases} -1, & j \leq s_\delta - \gamma \\ 0 & -\gamma < j < s_\delta + \gamma \\ 1 & j \geq s_\delta + \gamma \end{cases} \tag{28}$$

where $\gamma$ denotes threshold. The LTP feature is notified by $V_3$. Hence, the feature vector formed is stated by,

$$V = \{V_1, V_2, V_3\}. \tag{29}$$

## First-level classification to identify plant leaf type using TL with LeNet

After completing the feature extraction, the obtained feature vectors are applied for further inspection before being grouped into specific classes. The plant leaf type classification is an

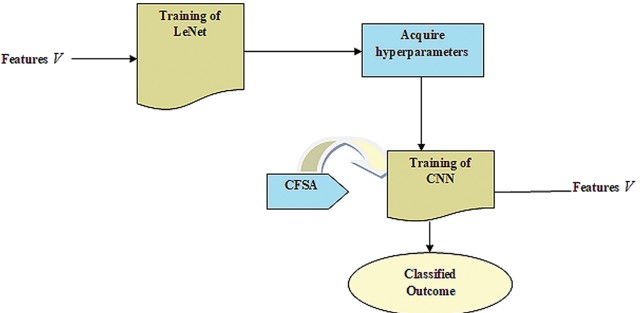

**Figure 2  LeNet with TL.**               

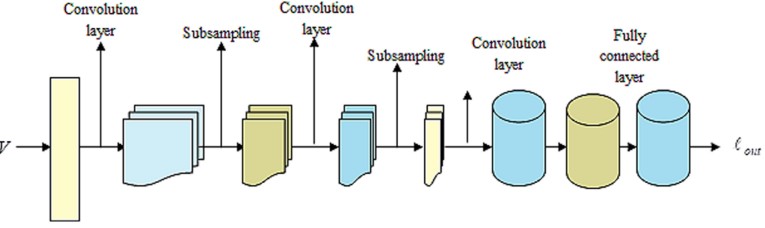

**Figure 3  Outlook of LeNet.**               

essential step in discovering the type of plant. In previous works, the researchers struggle and spend a huge time establishing the dataset by accumulating several leaf samples as raw databases. Here, the first level classification is considered which helps to recognize the type of plant leaf using TL with LeNet. The outlook of TL with LeNet is described along with CFSA steps.

### Outlook of TL with LeNet

The LeNet is simple and a minimum number of layers only required. Also, it needs less training time. TL is merged into a LeNet model to enhance the accuracy and model reliability when trained using a small quantity of data. Here, the Outlook of TL with LeNet is described herewith.

CNN gives enhanced efficacy with large datasets in contrast to smaller ones. It is useful in the applications of CNN wherein data tends to be small. Hence, the idea behind the TL is that it comprises a trained model with huge databases that are utilized for applications that contain small databases. Figure 2 provides LeNet with TL.

### Training of LeNet

LeNet (*Bouti et al., 2020*; *Wahlang et al., 2020*) indicates a current convolutional network that is devised for detecting plant leaf disease. It expresses a CNN having adequate input to generate several objects and various outputs. It can determine strings that lack prior segmentation. Hence, max pooling and layers of sparse convolution indicate the LeNet center. Thus, the lower layer contains the max pool and convolution layers. The output generated by LeNet is signified by $\ell_{out}$. Figure 3 illustrates a preview of the LeNet structure.

### Fetch hyperparameters

TL uses knowledge attained from the source to improve learning through the target area. In transferring parameters, the Hyperparameter acquired from the source is effectually utilized to optimize the target. It uses knowledge learned by source for improving the learning of the target. The LeNet model contains problems of low accuracy with few faulty data. Hence, acquiring knowledge from sources helps to improve LeNet's efficiency in the target. At last, the training instances of the target are utilized for performing parameters fine-tuning considering CNN-based transfer learning to fit the target.

### Training of CNN

The goal is to offer a CNN-based TL (*Shi et al., 2019*) with a pre-trained model like LeNet to classify the disease contained in plant leaves with improved accuracy.

## CNN model

CNN (*Aslam & Cui, 2020*) is well-known because of its enhanced efficiency in classifying the data. Here, the set of convolution layers and filters helps in extracting spatial and temporal features through data. The layer comprises a weight-sharing method which helps in reducing evaluations. The CNN expresses a feed forward ANN which comprises two problems.

### Steps of CFSA

The training of Transfer learning-based CNN with LeNet is done using CFSA. Here, the CFSA is obtained by inducing chronological concepts in FSA (*Zhiheng & Jianhua, 2021*). CFSA is motivated by the foraging and migratory characteristics of flamingos. It helps to fulfill global exploration as well as local exploitation abilities. Moreover, it is extensively competitive with classical techniques based on the speed of convergence, accuracy, and stability. FSA is utilized for finding and visualizing its optimization. It provides better solutions for optimum design in various research domains of optimization issues and offers novel solutions that can better help address these engineering design-related issues. The chronological idea is induced in FSA wherein the position data of the flamingo in the past iteration is adapted to produce the best solution. Based on this solution, the weights of neurons are updated using CFSA. Hence, the integration of chronological concept in FSA improves complete performance and the steps of CFSA are illustrated below:

#### Step 1) Initialization

The first chore is the initiation of solutions, and can be represented by,

$$J = \{J_1, J_2, \cdots, J_\varepsilon, \cdots, J_v\} \tag{30}$$

where, $v$ stands for total solutions, and $J_\varepsilon$ provides a $\varepsilon^{th}$ solution.

#### Step 2) Compute error

After initiating the solutions, the error of each solution is calculated. Thus, the Mean Square Error (MSE) is utilized and specified by,

$$Err = \frac{1}{e} \sum_{v=1}^{e} \left( Y_v^* - Y_v \right)^2 \tag{31}$$

Thus, $Y_v$ and expresses the original and predicted output of Transfer learning-based CNN with LeNet and $e$ signifies the total samples used.

**Step 3) Update using foraging behavior**

The moving task of flamingo forages in $l^{th}$ iteration and it is stated by,

$$n_{x,y}^l = \hat{\lambda}_1 \times in_y^l + I_2 \times \left| I_1 \times in_y^l + \hat{\lambda}_2 \times i_{x,y}^l \right| \tag{32}$$

The expression to update flamingo position using foraging behavior is given by,

$$i_{x,y}^{l+1} = \left( i_{x,y}^l + \hat{\lambda}_1 \times in_y^l + I_2 \times \left| I_1 \times in_y^l + \hat{\lambda}_2 \times i_{x,y}^l \right| \right) / U \tag{33}$$

where $i_{x,y}^{l+1}$ indicates the location of the $i^{th}$ flamingo in $j^{th}$ size and $(l+1)^{th}$ iteration, $i_{x,y}^l$ is the location of the $i^{th}$ flamingo in $j^{th}$ size and $l^{th}$ iteration, $in_y^l$ expresses the flamingo location with best fitness in $l^{th}$ iteration and $y^{th}$ size, $U$ is the diffusion factor, $I_1$ and $I_2$ depicts random numbers that undergo normal distribution and $\hat{\lambda}_2$ are randomized by $-1$ or 1.

**Step 4) Update using migrating behavior**

Whenever food is limited, the flamingos travel to subsequent regions where food is abundant. Consider that the food-rich position in $y^{th}$ dimension is $in_y$ and formulation of migration behavior of flamingo is stated as

$$i_{x,y}^{l+1} = i_{x,y}^l + X \times \left( in_y^l - i_{x,y}^l \right) \tag{34}$$

where, $X$ is Gaussian random number.

$$i_{x,y}^{l+1} = i_{x,y}^l + X \times in_y^l - X \times i_{x,y}^l \tag{35}$$

$$i_{x,y}^{l+1} = i_{x,y}^l[1 - X] + X \times in_y^l. \tag{36}$$

The above equation is rewritten as

$$i_{x,y}(l+1) = i_{x,y}(l)[1 - X] + X \times in_y(l). \tag{37}$$

At iteration $l$, the above expression is written as

$$i_{x,y}(l) = i_{x,y}(l-1)[1 - X] + X \times in_y(l-1). \tag{38}$$

Substitute Eq. (38) in Eq. (37),

$$i_{x,y}(l+1) = \left[ i_{x,y}(l-1)[1-X] + X \times in_y(l-1) \right][1-X] + X \times in_y(l) \tag{39}$$

$$i_{x,y}(l+1) = i_{x,y}(l-1)[1-X]^2 + X \times in_y(l-1)(1-X) + X \times in_y(l) \tag{40}$$

$$i_{x,y}(l+1) = i_{x,y}(l-1)[1-X]^2 + X\left[ in_y(l-1)(1-X) + in_y(l) \right] \tag{41}$$

Apply chronological concepts:

$$i_{x,y}(l+1) = \frac{i_{x,y}(l+1) + i_{x,y}(l+1)}{2}. \tag{42}$$

Substitute Eq. (37) and Eq. (41) in Eq. (42),

$$i_{x,y}(l+1) = \frac{i_{x,y}(l)[1-X] + X \times in_y(l) + i_{x,y}(l-1)[1-X]^2 + X\left[in_y(l-1)(1-X) + in_y(l)\right]}{2}. \tag{43}$$

***Step 5) Re-evaluate error***

The optimum solution is identified by calculating the error of newly generated position vectors and the position having the least error value is notified as the optimum solution.

***Step 6) Termination***

Steps are continued until the maximal iteration count is attained.

## Second-level classification to classify multi-class plant disease

Earlier detection and aversion of disease in plants are imperative factors in harvesting crops as they can effectually minimize any disorders of growth and hence reduce the application of pesticides for attaining better crop production. Hence, automatic classification of plant disease is an effective technique for attaining precision agriculture. Here, the second level classification involves the classification of multi-class plant diseases using TL with LeNet. Here, the tuning of TL with LeNet is done by CSFA.

## DISCUSSION OF OUTCOMES

Proficiency of CFSA+TL-based CNN+LeNet is produced by altering learning epoch and swarm size and it is examined using various kinds of criterions.

### Set-up of experiment

CFSA+TL-based CNN+LeNet is programmed in Python. The CFSA+TL-based CNN+LeNet parameters are presented in Table 1.

### Dataset description

The technique evaluation is performed with the Plant Village Dataset (*Mohanty, 2022*; https://github.com/spMohanty/PlantVillage-Dataset/tree/master/raw/color). It comprises 54,303 healthy and unhealthy images of the leaf which is split into 38 classes by species as well as disease. It is an open-access image repository that evaluates plant health to enable the design of mobile disease diagnosis. It is a dataset containing images of diseased plant leaf and their labels. There are 14 crop species available in the dataset, such as tomato, strawberry, squash, soy, raspberry, potato, pepper, peach, orange, grape, cherry, blueberry, and apple. There are 17 fundamental diseases, such as mold disease 2, mite disease 1, viral disease 2, and bacterial disease 4 in this dataset.

### Experimental upshots

Figure 4 shows experimental outputs of CFSA+TL-based CNN+LeNet using different images. The input image is explicated in Fig. 4A. The pre-processed image produced by adaptive anisotropic filtering is exposed in Fig. 4B. The segmented image obtained by U-

**Table 1 Parameter details.**

| Parameter | Value |
| --- | --- |
| Batch size | 128 |
| Epoch | 10 |
| Verbose | 0 |
| Learning rate | 0.5 |
| Loss | Categorical_crossentropy |
| Kernel size | (5, 5) |
| Optimizer | CFSA |
| Lower bound | 1 |
| Upper bound | 5 |
| Maximum iteration | 100 |

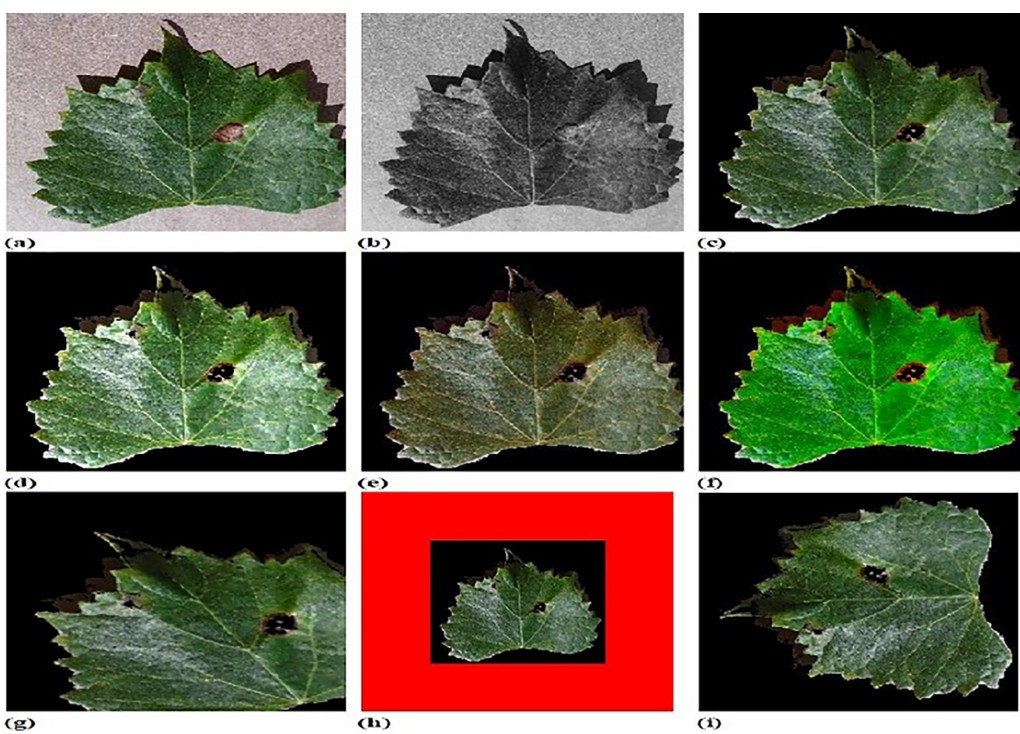

**Figure 4 Experimental outputs of CFSA+TL-based CNN+LeNet with (A) input image (B) pre-processed image (C) segmented image (D) contrast enhanced image (E) hue image (F) saturated image (G) affine transformed image (H) padded image (I) rotated image (J) translated image.**

Net++ is endowed in Fig. 4C. The contrast-enhanced image is depicted in Fig. 4D. The hue image is exposed in Fig. 4E. The saturated image obtained is depicted in Fig. 4F. The Affine transformed image is displayed in Fig. 4G. The padded image is depicted in Fig. 4H. The Rotated image is endowed in Fig. 4I. The translated image produced is explicated in Fig. 4J.

Figure 5 depicts the outputs of CFSA+TL-based CNN+LeNet with different image augmentation techniques along with extracted features. The affine transformation of

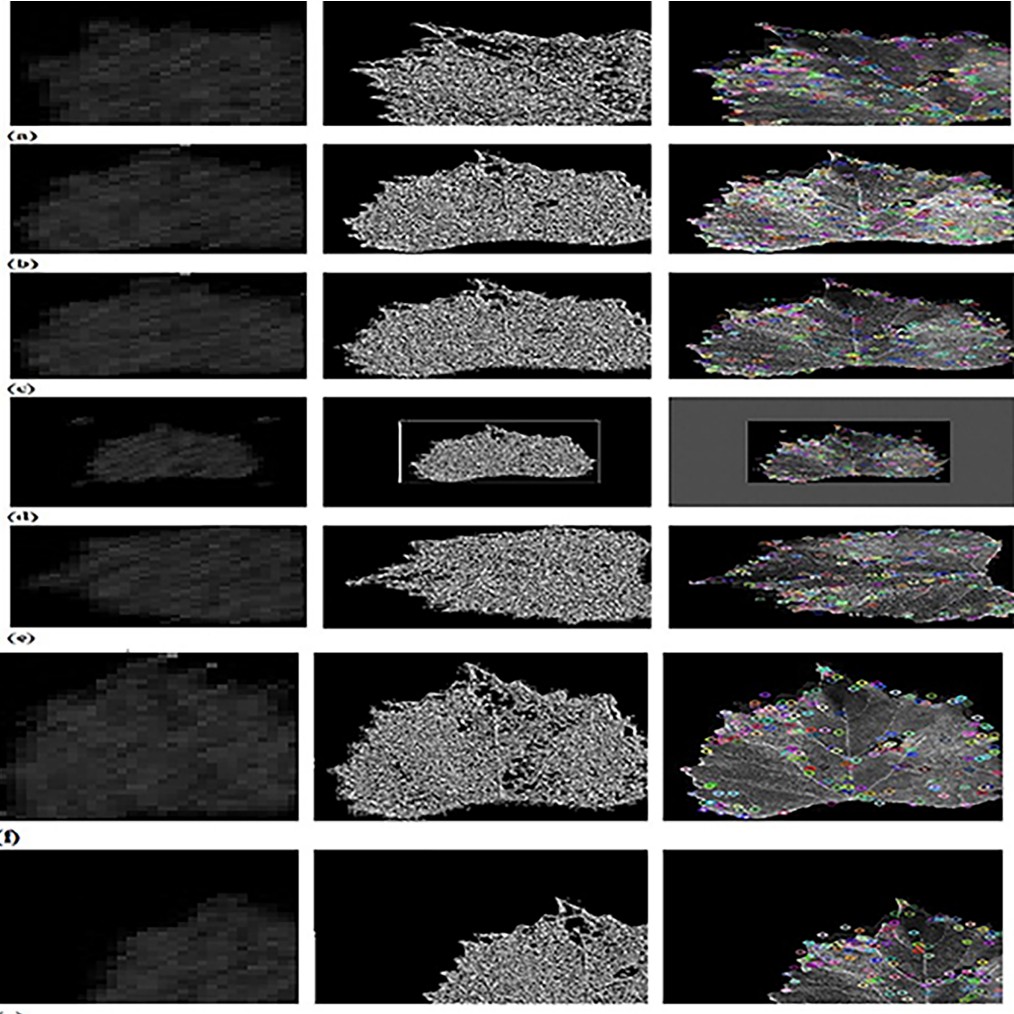

**Figure 5 Experimental outcomes of CFSA+TL-based CNN+LeNet with (A) affine transformation of DWT-OCLBP, LTP and SIFT (B) contrast of DWT-OCLBP image, LTP and SIFT (C) hue of DWT-OCLBP image, LTP and SIFT (D) padding of DWT-OCLBP image, LTP and SIFT (E) rotation of DWT-O.**

DWT-OCLBP, LTP, and SIFT is explicated in Fig. 5A. The contrast of DWT-OCLBP image, LTP, and SIFT is displayed in Fig. 5B. The hue of the DWT-OCLBP image, LTP, and SIFT is induced in Fig. 5C. The Padding of DWT-OCLBP image, LTP, and SIFT is notified in Fig. 5D. The Rotation of the DWT-OCLBP image, LTP, and SIFT is noted in Fig. 5E. The Saturation of the DWT-OCLBP image, LTP, and SIFT is noted in Fig. 5F. The Translation of DWT-OCLBP image, LTP, and SIFT is noted in Fig. 5G.

## Metrics used

Proficiency of each scheme ability is observed by inspecting CFSA+TL-based CNN+LeNet with different performance parameters which is explained below.

### (a) Accuracy

It is considered as a metric of closeness degree to its true value and can be expressed as

$$M = \frac{B + V}{B + V + F + D}.\tag{44}$$

Hence, $V$ states true positive, $B$ offers true negative, $D$ gives false positive, and displays false negative.

### (b) Sensitivity

It depicts the proportion of true positives number to the total number of positives, and is presented by

$$N = \frac{V}{V + F}.\tag{45}$$

### (c) Specificity

It defines the ratio of negatives and is accurately identified and it is notated by

$$K = \frac{B}{B + D}.\tag{46}$$

### (d) F-measure

It is the compromise between precision and recall and it is noted

$$FM = \frac{V}{V + \frac{1}{2}(D + F)}.\tag{47}$$

## Algorithm methods

The algorithm efficiency estimation considered for analysis is PSO+TL-based CNN+LeNet (*Wang, Tan & Liu, 2018*; *Bouti et al., 2020*), CSO+TL-based CNN+LeNet (*Cheng & Jin, 2014*; *Bouti et al., 2020*), ROA+TL-based CNN+LeNet (*Jia, Peng & Lang, 2021*; *Bouti et al., 2020*), GTO+TL-based CNN+LeNet (*Xiao et al., 2022*; *Bouti et al., 2020*), FSA+TL-based CNN+LeNet (*Zhiheng & Jianhua, 2021*; *Bouti et al., 2020*), and proposed CFSA+TL-based CNN+LeNet.

## Algorithmic analysis

The evaluation of scheme efficacy with first-level and second-level classifications is described with different metrics by altering swarm size along the x-axis.

### a) Graphical estimation of algorithm efficacy with first-level classification

Figure 6 gives an evaluation of algorithm efficacy with first-level classification considering different metrics. The graph depicting accuracy is explicated in Fig. 6A. When the swarm's size is 20, the accuracy produced by PSO+TL-based CNN+LeNet is 0.837, CSO+TL-based CNN+LeNet is 0.865, ROA+TL-based CNN+LeNet is 0.887, GTO+ TL-based CNN+LeNet is 0.898, FSA+TL-based CNN+LeNet is 0.918, and CFSA+TL-based CNN+LeNet is 0.946. The graph denoting analysis considering sensitivity is explicated in

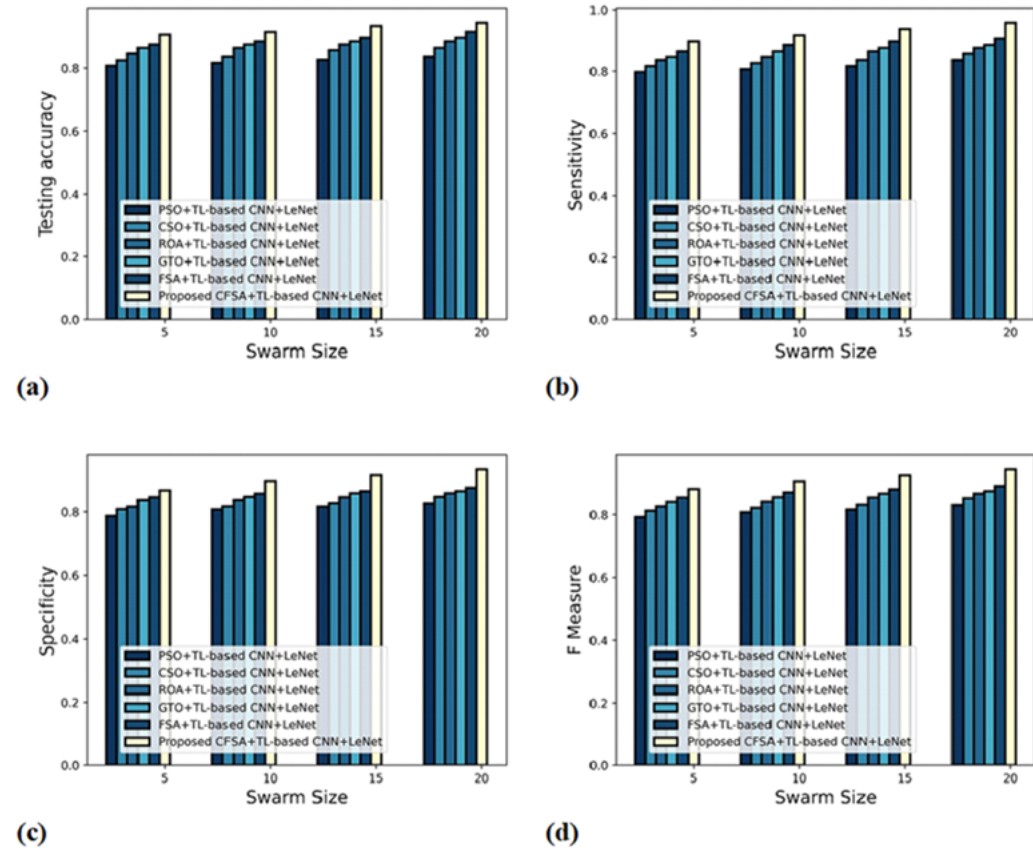

**Figure 6 Evaluation of algorithm efficacy with first level classification considering (A) accuracy (B) sensitivity (C) specificity (D) F-measure.**

Fig. 6B. For swarms size is 20, the elevated sensitivity of 0.958 is produced by CFSA+TL-based CNN+LeNet whilst sensitivity of PSO+TL-based CNN+LeNet, CSO+TL-based CNN+LeNet, ROA+TL-based CNN+LeNet, GTO+TL-based CNN+LeNet, FSA+TL-based CNN+LeNet are 0.837, 0.858, 0.877, 0.887, and 0.908. The graph regarding specificity analysis is considered in Fig. 6C. Considering swarms size as 20, the specificity generated is 0.827 for PSO+TL-based CNN+LeNet, 0.848 for CSO+TL-based CNN+LeNet, 0.859 for ROA+TL-based CNN+LeNet, 0.865 for GTO+TL-based CNN+LeNet, 0.877 for FSA+TL-based CNN+LeNet, and 0.936 for CFSA+TL-based CNN+LeNet. The graph regarding F-Measure analysis is considered in Fig. 6D. Considering swarms size as 10, the F-measure generated is 0.808 for PSO+TL-based CNN+LeNet, 0.822 for CSO+TL-based CNN +LeNet, 0.842 for ROA+TL-based CNN+LeNet, 0.857 for GTO+TL-based CNN+LeNet, 0.872 for FSA+TL-based CNN+LeNet, and 0.907 for CFSA+TL-based CNN+LeNet.

*b) Graphical evaluation of algorithm efficacy with second-level classification*

Figure 7 gives the evaluation of algorithm efficacy with second-level classification considering different metrics. The graph depicting accuracy analysis is explicated in Fig. 7A. When swarms size is 20, the accuracy noted by PSO+TL-based CNN+LeNet is 0.877, CSO+ TL-based CNN+LeNet is 0.887, ROA+TL-based CNN+LeNet is 0.898, GTO

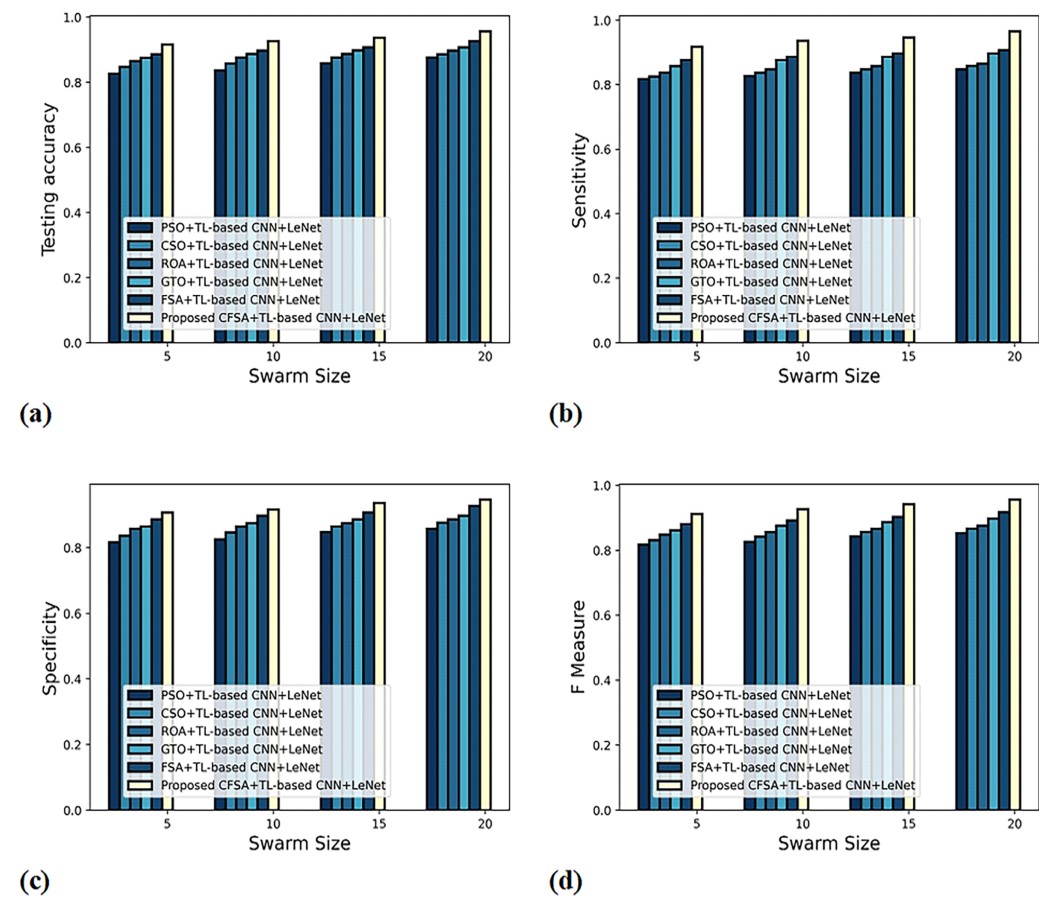

**Figure 7 Evaluation of algorithm efficacy with second level classification with (A) accuracy (B) sensitivity (C) specificity (D) F-measure.**

+TL-based CNN+LeNet is 0.908, FSA+TL-based CNN+LeNet is 0.927, and CFSA+TL-based CNN+LeNet is 0.957. The graph denoting analysis considering sensitivity is explicated in Fig. 7B. When swarms size is 20, the elevated sensitivity of 0.965 is produced by CFSA+TL-based CNN+LeNet whilst sensitivity of PSO+TL-based CNN+LeNet, CSO+TL-based CNN+LeNet, ROA+TL-based CNN+LeNet, GTO+TL-based CNN+LeNet, FSA+ TL-based CNN+LeNet are 0.848, 0.858, 0.865, 0.898, 0.908. The graph regarding specificity analysis is considered in Fig. 7C. For swarms size is 20, the specificity produced is 0.858 for PSO+ TL-based CNN+LeNet, 0.877 for CSO+TL-based CNN+LeNet, 0.887 for ROA+TL-based CNN+LeNet, 0.898 for GTO+TL-based CNN+LeNet, 0.928 for FSA+TL-based CNN+LeNet, and 0.947 for CFSA+TL-based CNN+LeNet. The graph regarding F-Measure analysis is considered in Fig. 7D. For swarms size is 10, the F-measure produced is 0.826 for PSO+ TL-based CNN+LeNet, 0.842 for CSO+TL-based CNN+LeNet, 0.856 for ROA+TL-based CNN+LeNet, 0.876 for GTO+TL-based CNN+LeNet, 0.892 for FSA+TL-based CNN+LeNet, and 0.927 for CFSA+TL-based CNN+LeNet.

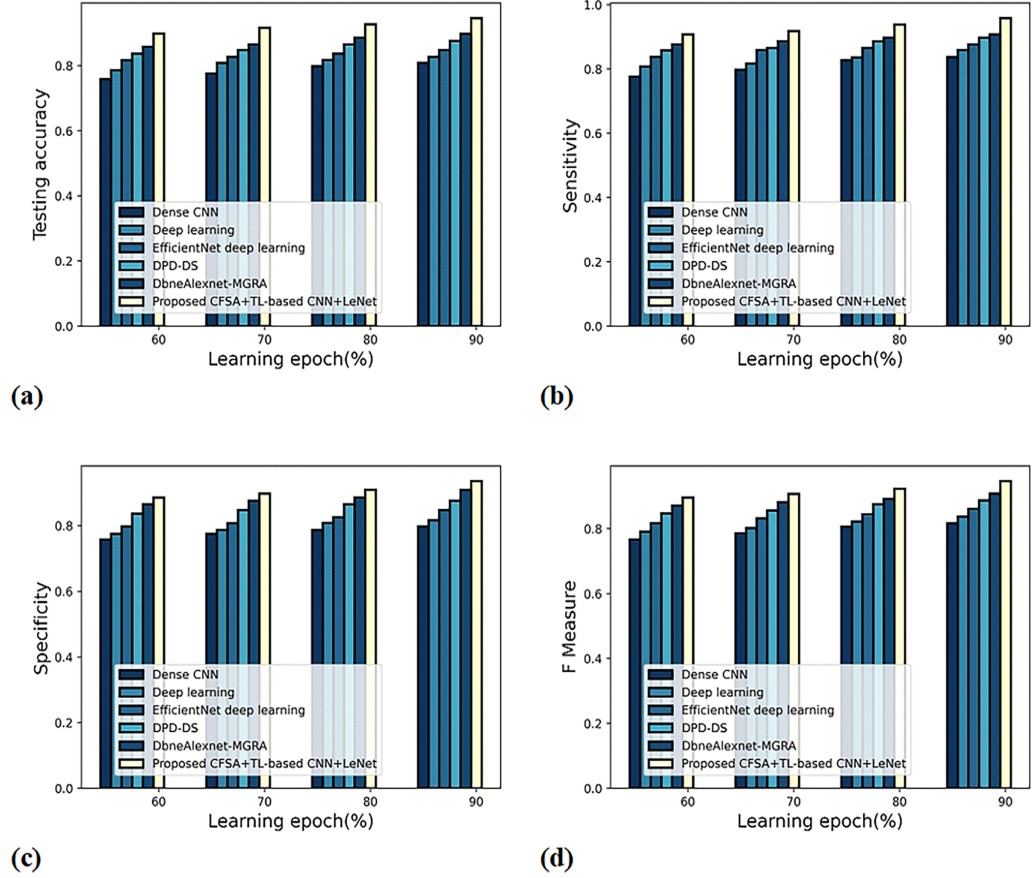

**Figure 8 Evaluation of scheme efficacy with first level classification considering (A) accuracy (B) sensitivity (C) specificity (D) F-measure.**

## Comparative methods

The schemes efficiency estimation adapted for assessment are dense CNN (*Tiwari, Joshi & Dutta, 2021*), deep learning (*Roy & Bhaduri, 2021*), EfficientNet deep learning (*Atila et al., 2021*), DPD-DS (*Lakshmi & Savarimuthu, 2021*), DbneAlexnet-MGRA, and CFSA+TL-based CNN+LeNet.

## Comparative analysis

The valuation of technique efficacy with first and second-level categorization is defined with various measures by altering the learning epoch along the x-axis.

*a) Graphical estimation of scheme efficacy with first-level categorization*

Figure 8 gives an evaluation of scheme efficacy with first-level classification considering different metrics. The graph depicting accuracy analysis is noted in Fig. 8A. When the learning epoch is 90%, the accuracy produced by dense CNN is 0.809, Deep learning is 0.827, EfficientNet deep learning is 0.848, DPD-DS is 0.877, DbneAlexnet-MGRA is 0.898, and CFSA+TL-based CNN+LeNet is 0.946. The graph denoting analysis considering sensitivity is noted in Fig. 8B. When the learning epoch is 90%, the sensitivity produced by

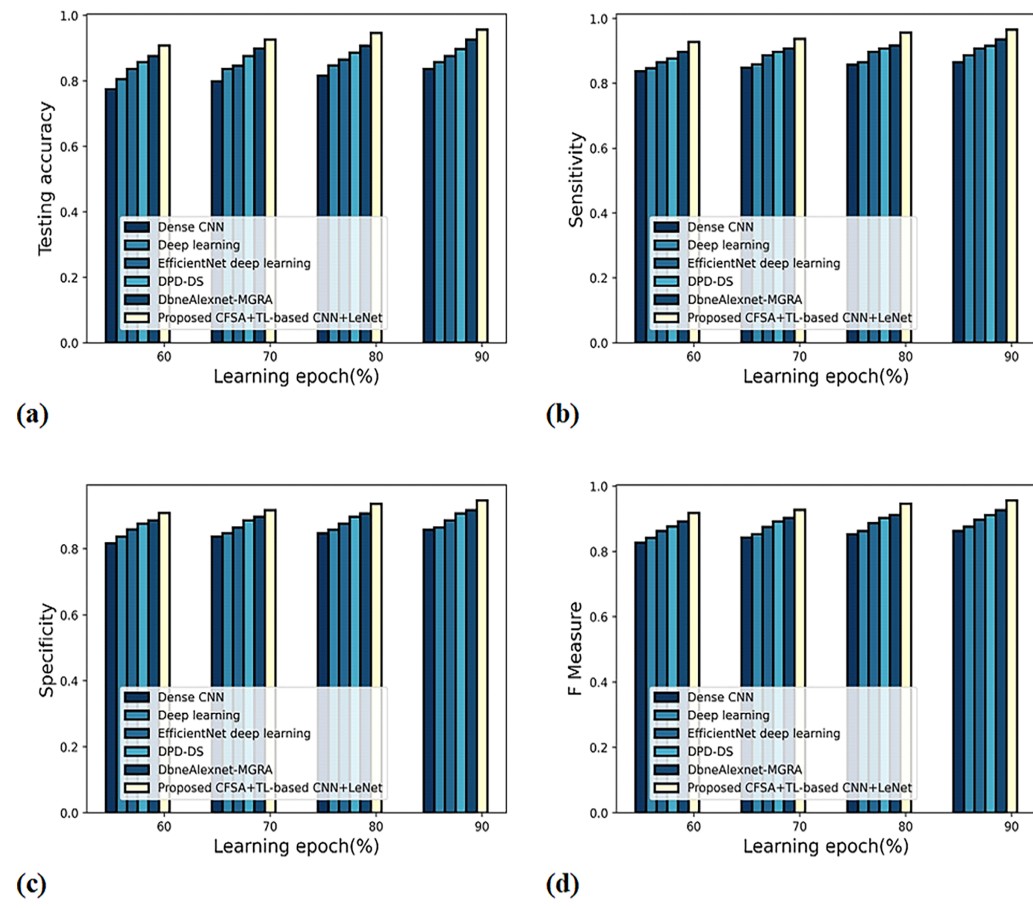

**Figure 9** Evaluation of scheme efficacy with second level categorization considering (A) accuracy (B) sensitivity (C) specificity (D) F-measure. 

dense CNN is 0.837, Deep learning is 0.859, EfficientNet deep learning is 0.877, DPD-DS is 0.898, DbneAlexnet-MGRA is 0.908, and CFSA+TL-based CNN+LeNet is 0.958. The graph regarding specificity analysis is considered in Fig. 8C. When the learning epoch is 90%, the specificity produced is 0.798 for dense CNN, 0.817 for deep learning, 0.848 for EfficientNet deep learning, 0.877 for DPD-DS, 0.909 for DbneAlexnet-MGRA, and 0.936 for CFSA+TL-based CNN+LeNet. The graph regarding F-measure analysis is considered in Fig. 8D. When the learning epoch is 90%, the specificity produced is 0.817 for dense CNN, 0.837 for deep learning, 0.862 for EfficientNet deep learning, 0.887 for DPD-DS, 0.908 for DbneAlexnet-MGRA, and 0.947 for CFSA+TL-based CNN+LeNet.

*b) Graphical evaluation of scheme efficacy with second-level categorization*

Figure 9 gives the evaluation of scheme efficacy using different metrics. The graph depicting accuracy is noted in Fig. 9A. When the learning epoch is 90%, the accuracy produced by dense CNN is 0.837, Deep learning is 0.858, EfficientNet deep learning is 0.877, DPD-DS is 0.898, DbneAlexnet-MGRA is 0.927, and CFSA+TL-based CNN+LeNet is 0.957. The graph denoting analysis considering sensitivity is explicated in Fig. 9B. When the learning epoch is 90%, the elevated sensitivity of 0.965 is produced by CFSA+TL-based CNN+LeNet whilst the sensitivity of the remaining schemes is 0.865, 0.887, 0.908, 0.916,

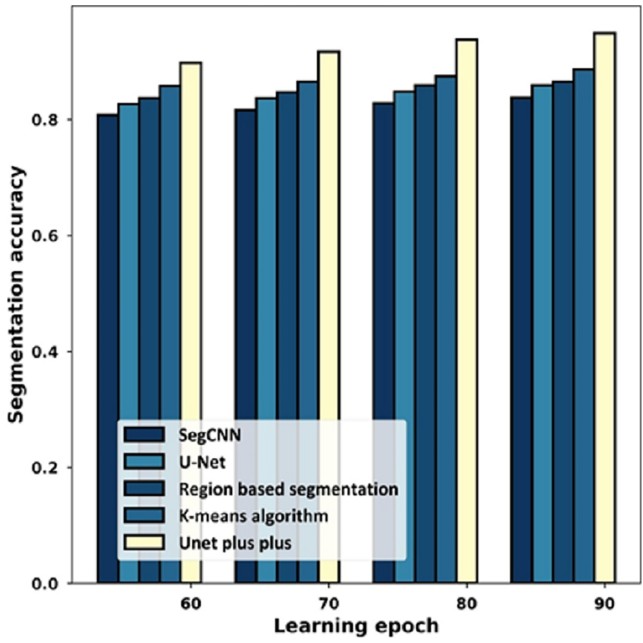

**Figure 10 Estimation with segmentation accuracy.**

and 0.936. The graph regarding specificity analysis is considered in Fig. 9C. When the learning epoch is 90%, the specificity produced is 0.859 for dense CNN, 0.865 for deep learning, 0.887 for EfficientNet deep learning, 0.908 for DPD-DS, 0.918 for DbneAlexnet-MGRA, and 0.947 for CFSA+TL-based CNN+LeNet. The graph regarding F-measure analysis is considered in Fig. 9D. When the learning epoch is 90%, the F-measure produced is 0.862 for dense CNN, 0.876 for deep learning, 0.897 for EfficientNet deep learning, 0.912 for DPD-DS, 0.927 for DbneAlexnet-MGRA, and 0.956 for CFSA+TL-based CNN+LeNet.

### Analysis using segmentation accuracy

Figure 10 provides the estimation with the segmentation accuracy metric. For the 60% learning epoch, the segmentation accuracy noted by SegCNN is 0.808, U-Net is 0.827, Region-based segmentation is 0.837, K-mean algorithm is 0.858, and U-Net++ is 0.898. Considering the 90% learning epoch, the high segmentation accuracy of 0.949 is noted by U-Net++ whilst the segmentation accuracy of the remaining schemes is 0.838, 0.859, 0.865, 0.887.

### Comparative estimate

The efficiency of schemes and algorithms considering different metrics are described below.

### Algorithm estimate

Table 2 defines algorithm efficiency assessment considering diverse performance criteria. Considering first-level classification, the augmented accuracy of 94.6% is observed by CFSA+TL-based CNN+LeNet while the accuracy of enduring schemes is 83.7%, 86.5%, 88.7%, 89.8%, and 91.8%. The high sensitivity of 95.8% is produced by CFSA+TL-based

**Table 2 Algorithm efficacy evaluation.**

| Level | Metrics | PSO+LeNet | CSO+LeNet | ROA+LeNet | GTO+LeNet | FSA+LeNet | CFSA+TL-based CNN+LeNet |
|-------|---------|-----------|-----------|-----------|-----------|-----------|--------------------------|
| First level | Accuracy (%) | 83.7 | 86.5 | 88.7 | 89.8 | 91.8 | **94.6** |
| | Sensitivity (%) | 83.7 | 85.8 | 87.7 | 88.7 | 90.8 | **95.8** |
| | Specificity (%) | 82.7 | 84.8 | 85.9 | 86.5 | 87.7 | **93.6** |
| | F-measure (%) | 83.2 | 85.3 | 86.8 | 87.6 | 89.2 | **94.7** |
| Second level | Accuracy (%) | 87.7 | 88.7 | 89.8 | 90.8 | 92.7 | **95.7** |
| | Sensitivity (%) | 84.8 | 85.8 | 86.5 | 89.8 | 90.8 | **96.5** |
| | Specificity (%) | 85.8 | 87.7 | 88.7 | 89.8 | 92.8 | **94.7** |
| | F-measure (%) | 85.3 | 86.7 | 87.6 | 89.8 | 91.7 | **95.6** |

**Note:**
The best results are shown in bold.

**Table 3 Scheme efficacy evaluation.**

| Level | Metrics | Dense CNN | Deep learning | EfficientNet deep learning | DPD-DS | DbneAlexnet-MGRA | CFSA+TL-based CNN +LeNet |
|-------|---------|-----------|---------------|----------------------------|--------|------------------|--------------------------|
| First level | Accuracy (%) | 80.9 | 82.7 | 84.8 | 87.7 | 89.8 | **94.6** |
| | Sensitivity (%) | 83.7 | 85.9 | 87.7 | 89.8 | 90.8 | **95.8** |
| | Specificity (%) | 79.8 | 81.7 | 84.8 | 87.7 | 90.9 | **93.6** |
| | F-measure (%) | 81.7 | 83.7 | 86.2 | 88.7 | 90.8 | **94.7** |
| Second level | Accuracy (%) | 83.7 | 85.8 | 87.7 | 89.8 | 92.7 | **95.7** |
| | Sensitivity (%) | 86.5 | 88.7 | 90.8 | 91.6 | 93.6 | **96.5** |
| | Specificity (%) | 85.9 | 86.5 | 88.7 | 90.8 | 91.8 | **94.7** |
| | F-measure (%) | 86.2 | 87.6 | 89.7 | 91.2 | 92.7 | **95.6** |

**Note:**
The best results are shown in bold.

CNN+LeNet while the sensitivity of enduring schemes is 83.7%, 85.8%, 87.7%, 88.7%, and 90.8%. The high specificity of 93.6% is produced by CFSA+TL-based CNN+LeNet while the specificity of enduring schemes is 82.7%, 84.8%, 85.9%, 86.5%, and 87.7%. The highest F-measure is 94.7%. Considering second-level classification, the augmented accuracy of 95.7%, sensitivity of 96.5%, specificity of 94.7%, and F-measure of 95.6% is noted by CFSA +TL-based CNN+LeNet. From the evaluation, it is observed that the CFSA+TL-based CNN+LeNet has improved its ability to provide a better classification of plant leaf disease.

## Scheme evaluation

Table 3 describes the scheme efficiency computation with various evaluation criteria. Considering first-level classification, the augmented accuracy of 94.6% is observed by CFSA+TL-based CNN+LeNet while the accuracy of enduring schemes is 80.9%, 82.7%, 84.8%, 87.7%, and 89.8%. The high sensitivity of 95.8% is noted by CFSA+TL-based CNN +LeNet whilst the sensitivity of enduring schemes is 83.7%, 85.9%, 87.7%, 89.8%, and 90.8%. The high specificity of 93.6% is produced by CFSA+TL-based CNN+LeNet whilst the specificity of enduring schemes is 79.8%, 81.7%, 84.8%, 87.7%, and 90.9%. The high F-measure of 94.7% is produced by CFSA+TL-based CNN+LeNet whilst the F-measure of enduring schemes are 81.7%, 83.7%, 86.2%, 88.7%, and 90.8%. Considering second-level

**Table 4 Statistical analysis.**

| Methods | Accuracy | | | Sensitivity | | | Specificity | | | F-measure | | |
|---|---|---|---|---|---|---|---|---|---|---|---|---|
| | Best | Mean | Variance | Best | Mean | Variance | Best | Mean | Variance | Best | Mean | Variance |
| First level | | | | | | | | | | | | |
| Dense CNN | 0.809 | 0.805 | 0.004 | 0.837 | 0.834 | 0.003 | 0.798 | 0.794 | 0.004 | 0.817 | 0.813 | 0.004 |
| Deep learning | 0.827 | 0.824 | 0.003 | 0.859 | 0.856 | 0.003 | 0.817 | 0.812 | 0.005 | 0.837 | 0.834 | 0.003 |
| EfficientNet deep learning | 0.848 | 0.845 | 0.003 | 0.877 | 0.873 | 0.004 | 0.848 | 0.845 | 0.003 | 0.862 | 0.859 | 0.003 |
| DPD-DS | 0.877 | 0.874 | 0.003 | 0.898 | 0.896 | 0.002 | 0.877 | 0.875 | 0.002 | 0.887 | 0.883 | 0.004 |
| DbneAlexnet-MGRA | 0.898 | 0.895 | 0.003 | 0.908 | 0.905 | 0.003 | 0.909 | 0.905 | 0.004 | 0.908 | 0.904 | 0.004 |
| CFSA+TL-based CNN+LeNet | 0.946 | 0.944 | 0.002 | 0.958 | 0.957 | 0.001 | 0.936 | 0.934 | 0.002 | 0.947 | 0.945 | 0.002 |
| Second level | | | | | | | | | | | | |
| Dense CNN | 0.837 | 0.834 | 0.003 | 0.865 | 0.860 | 0.005 | 0.859 | 0.855 | 0.004 | 0.862 | 0.860 | 0.002 |
| Deep learning | 0.858 | 0.855 | 0.003 | 0.887 | 0.884 | 0.003 | 0.865 | 0.861 | 0.004 | 0.876 | 0.873 | 0.003 |
| EfficientNet deep learning | 0.877 | 0.874 | 0.003 | 0.908 | 0.904 | 0.004 | 0.887 | 0.884 | 0.003 | 0.897 | 0.893 | 0.004 |
| DPD-DS | 0.898 | 0.894 | 0.004 | 0.916 | 0.913 | 0.003 | 0.908 | 0.904 | 0.004 | 0.912 | 0.910 | 0.002 |
| DbneAlexnet-MGRA | 0.927 | 0.924 | 0.003 | 0.936 | 0.933 | 0.003 | 0.918 | 0.916 | 0.002 | 0.927 | 0.924 | 0.003 |
| CFSA+TL-based CNN+LeNet | 0.957 | 0.955 | 0.002 | 0.965 | 0.963 | 0.002 | 0.947 | 0.946 | 0.001 | 0.956 | 0.955 | 0.001 |

classification, the augmented accuracy of 95.7%, sensitivity of 96.5%, specificity of 94.7%, and F-measure of 95.6% is noted by CFSA+TL-based CNN+LeNet. From the analysis, it is revealed that the CFSA+TL-based CNN+LeNet is effective in performing the classification to find and classify diseases present in plant leaves with improved accuracy.

### Statistical analysis

Table 4 describes the statistical analysis of the CFSA+TL-based CNN+LeNet. It is based on the best, mean, and variance of the model using various evaluation metrics.

## CONCLUSION

The aim is to design a framework with leaf images using CFSA with TL-based CNN and the LeNet model. Initially, the leaf image is acquired and offered to adaptive anisotropic diffusion for filtering noise. Thereafter the segmentation of plant leaf is done with U-Net++ and trained using MGRA. After this, the image augmentation is done which is classified into Position augmentation like padding, rotation, translation, affine transformation, and color augmentation including contrast, saturation, and hue. Then the feature extraction is done by using Hybrid OCLBP-based DWT with HoG, SIFT, and LTP. Next, the first level of classification is considered as plant type classification and the second level of categorization is considered as the multi-class plant disease classification, and both classifications are done by using a CFSA with TL-based CNN and LeNet model. The proposed CFSA-TL-based CNN with LeNet outperformed with a high accuracy of 95.7%, a sensitivity of 96.5%, and a specificity of 94.7%. The goal is to optimize the features, reduce computational time, and produce better accuracy in detecting and classifying plant leaf disease. The proposed method is used to identify and classify the disease in its early stage,

which elevates the productivity and quality of food at less cost and more profit. However, the limited dataset is used in this research for the analysis. In the future, other datasets will be applied to reveal the reliability and performance of the designed scheme.

### Funding
The authors received no funding for this work.

### Competing Interests
The authors declare that they have no competing interests.

### Author Contributions
- Malathi Chilakalapudi conceived and designed the experiments, performed the experiments, performed the computation work, prepared figures and/or tables, and approved the final draft.
- Sheela Jayachandran analyzed the data, authored or reviewed drafts of the article, and approved the final draft.

### Data Availability
The data are available at GitHub: https://github.com/spMohanty/PlantVillage-Dataset/tree/master/raw/color (Sharada Mohanty).

The code is available in the Supplemental File.

### Supplemental Information
Supplemental information for this article can be found online at http://dx.doi.org/10.7717/peerj-cs.1972#supplemental-information.

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
