# Peer review of "Multi-classification of disease induced in plant leaf using chronological Flamingo search optimization with transfer learning"

_PeerJ Computer Science, doi:10.7717/peerj-cs.1972_

## Round 0.1 · original submission · Major Revisions

Dear authors,

Thank you for submitting your article. The reviewers’ comments are now available. Your article has not been recommended for publication in its current form. However, we encourage you to address the reviewers' concerns and criticisms; particularly regarding readability, quality, experimental design and validity, and resubmit your article once you have updated it accordingly.

Reviewers have requested that you may cite specific references. You may add them if you believe they are especially relevant and useful. However, I do not expect you to include these citations, and if you do not include them, this will not influence my decision.

Furthermore, the title of your paper is not consistent with the performed work. There is not any "Chronological avian search optimization" algorithm in your paper. It is also not used in the literature. This confusion should be corrected.

Best wishes,

**Language Note:** The review process has identified that the English language must be improved. PeerJ can provide language editing services - please contact us at copyediting@peerj.com for pricing (be sure to provide your manuscript number and title). Alternatively, you should make your own arrangements to improve the language quality and provide details in your response letter. – PeerJ Staff

·

Basic reporting

The provided text appears to be a computer science research paper focused on the application of a novel algorithm, Chronological Flamingo Search Algorithm (CFSA), in conjunction with transfer learning for the multi-classification of plant leaf diseases using convolutional neural networks (CNNs) with LeNet architecture. The paper also involves pre-processing steps, segmentation of plant leaves using U-Net++, and various image augmentation techniques.

However, there are several issues related to basic reporting:

1. **Organization and Formatting:** The text lacks proper organization and formatting. Sections and subsections are not clearly defined, making it challenging to follow the flow of the paper. There are also inconsistencies in numbering and indentation.

2. **Figures and Tables:** Figure captions are missing, and it seems that Figure 1 is referenced but not included in the provided text. The absence of visual representations makes it difficult to understand the proposed model.

3. **References:** The references are not complete, and some entries are incomplete or incorrectly formatted. Proper citation styles and consistent formatting should be ensured.
Consider add these references
1. Subramanian, M., Lv, N. P., & VE, S. (2022). Hyperparameter optimization for transfer learning of VGG16 for disease identification in corn leaves using Bayesian optimization. Big Data, 10(3), 215-229.
2. Krishnamoorthy, N., Prasad, L. N., Kumar, C. P., Subedi, B., Abraha, H. B., & Sathishkumar, V. E. (2021). Rice leaf diseases prediction using deep neural networks with transfer learning. Environmental Research, 198, 111275.
3. Sathishkumar, V. E., Rahman, A. B. M., Park, J., Shin, C., & Cho, Y. (2020, April). Using machine learning algorithms for fruit disease classification. In Basic & clinical pharmacology & toxicology (Vol. 126, pp. 253-253). 111 RIVER ST, HOBOKEN 07030-5774, NJ USA: WILEY.
4. Subramanian, M., Sathishkumar, V. E., Cho, J., & Shanmugavadivel, K. (2023). Learning without forgetting by leveraging transfer learning for detecting COVID-19 infection from CT images. Scientific Reports, 13(1), 8516.

4. **Language and Grammar:** The text contains numerous grammatical errors and awkward sentences, impacting the clarity of communication. The language needs to be refined for better readability.

5. **Redundancy:** Some information, especially in the introduction, is repeated, leading to redundancy. The paper should maintain conciseness and avoid unnecessary repetition.

6. **Abstract:** The abstract provides a general overview but lacks specific details about the methodology, results, and conclusions. An abstract should briefly summarize each section of the paper.

In order to improve the basic reporting, the author should address these issues by reorganizing the content, providing proper figure captions, correcting references, improving language and grammar, eliminating redundancy, and enhancing the abstract's informativeness. Additionally, it would be beneficial to include the missing Figure 1 and any other relevant figures or tables to aid in understanding the proposed model.

Experimental design

The experimental design section of the paper details the methodology employed for evaluating the proposed Chronological Flamingo Search Algorithm (CFSA) and transfer learning in the context of plant leaf disease multi-classification. However, there are several issues with the experimental design:

1. **Clarity in Methodology:** The methodology lacks clarity and specificity. It is crucial to provide detailed steps and parameters used in each stage, including pre-processing, segmentation, and the application of the CFSA algorithm.

2. **Dataset Description:** While the dataset is mentioned, there is a lack of detailed information about its source, size, and characteristics. A comprehensive understanding of the dataset is essential for the readers to evaluate the experiment's validity.

3. **Baseline Comparison:** The paper lacks a clear comparison with existing or baseline methods. To demonstrate the effectiveness of the proposed approach, a comparison with other state-of-the-art methods or standard algorithms is necessary.

4. **Evaluation Metrics:** The choice of evaluation metrics needs to be justified. If metrics such as accuracy, precision, recall, or F1 score are used, the reasons for selecting these specific metrics should be explained.

5. **Parameters and Hyperparameters:** The values of parameters and hyperparameters used in the experiments are not provided. It is essential to disclose these details for reproducibility and to understand the robustness of the proposed method.

6. **Randomness Control:** If randomness is involved in any part of the algorithm or experiments, the paper should discuss how randomness is controlled to ensure consistent and reproducible results.

7. **Statistical Analysis:** The absence of statistical analysis or significance testing is a limitation. Statistical tests can strengthen the validity of the experimental results.

8. **Discussion of Limitations:** There is no discussion of potential limitations in the experimental design. Addressing the limitations can provide a more comprehensive view of the proposed method's applicability.

To enhance the experimental design, the author should provide more detailed and specific information on the methodology, dataset, baseline comparisons, evaluation metrics, parameters, randomness control, statistical analysis, and limitations. This will contribute to a more robust and transparent evaluation of the proposed algorithm.

Validity of the findings

The validity of the findings is a critical aspect of any scientific research. In the context of the paper on the Chronological Flamingo Search Algorithm (CFSA) for plant leaf disease multi-classification using transfer learning, the following points need attention to ensure the validity of the findings:

1. **Reproducibility:** The paper lacks information on how the experiments can be reproduced. It is important to provide clear instructions or code availability to enable other researchers to replicate the results independently.

2. **Experimental Rigor:** The paper should discuss the experimental rigor, including the number of experiments conducted and variations tested. A robust experimental setup with multiple runs and different scenarios can enhance the reliability of the findings.

3. **Discussion of Outliers:** If there are outliers or unexpected results, the paper should address them and provide possible explanations. Ignoring outliers without explanation may raise concerns about the reliability of the findings.

4. **Statistical Significance:** If applicable, statistical tests should be employed to determine the significance of the observed results. This is crucial for establishing the reliability of any improvements claimed by the CFSA algorithm compared to baseline methods.

5. **Generalization:** The paper needs to discuss the generalization of findings to different datasets or scenarios. It should be clear whether the proposed algorithm is specific to the dataset used or if it has broader applicability.

6. **Comparison with Baselines:** The paper should explicitly compare the performance of the CFSA algorithm with baseline methods or existing state-of-the-art algorithms. This comparative analysis is essential for evaluating the novelty and effectiveness of the proposed approach.

7. **Discussion of Limitations:** Addressing the limitations of the study is important for establishing the scope and applicability of the proposed algorithm. This includes discussing scenarios where the CFSA algorithm may not perform optimally.

8. **Ethical Considerations:** Depending on the nature of the research, ethical considerations related to data usage, experimental design, and potential biases should be addressed.

To enhance the validity of the findings, the author should provide details on reproducibility, experimental rigor, handling outliers, statistical significance, generalization, baseline comparisons, limitations, and ethical considerations. These considerations will contribute to a more robust and trustworthy scientific contribution.

Additional comments

Figures and Tables: Evaluate the quality and relevance of figures and tables. Check if they are properly labeled, cited in the text, and contribute effectively to the understanding of the research. Suggest improvements if needed.

Data Presentation: Assess how well the data is presented. Ensure that data is accurate, relevant, and appropriately analyzed. If there are opportunities to enhance data visualization or interpretation, provide constructive feedback.

Conclusion and Implications: Examine the conclusion section to ensure that it effectively summarizes the key findings and discusses their implications. Verify that the conclusion aligns with the objectives set in the introduction.

Future Work: If applicable, review the section on future work. Ensure that it provides meaningful insights into potential extensions or applications of the current research. Evaluate whether the proposed future directions are logical and well-founded.

Overall Contribution: Assess the overall contribution of the paper to the field. Determine whether the research provides a meaningful advancement in knowledge or methodology. Clarify the unique aspects that distinguish this work from existing literature.

Reviewer 2 ·

Basic reporting

1-In the abstract section of the article, the "c" in the phrase "convolutional Neural Network (CNN)" should be written with a capital letter.
2- In the abstract section of the article, the "a" in the phrase "Chronological Flamingo Search algorithm (CFSA)" should be written with a capital letter.
3-In the abstract of the article, the authors state, "The CFSA-TL-based CNN with LeNet provided better accuracy of 95.7%, sensitivity of 96.5%, and specificity of 94.7%." They wrote the sentence. While it is a known fact that LeNet is an extremely old CNN architecture, it should be explained why this architecture was chosen.
4-The motivation for the study needs to be given in more detail for the reader to understand.
5-It is recommended that this sentence be removed from the article. "India represents a country that has fame in the domain of agriculture in which the majority of the population relies on agriculture."
6-Give references to prove this sentence. "The majority of studies revealed that the quality of products can be reduced because of the diseases that occur in plants." Studies such as "Hybrid Deep Model for Automated Detection of Tomato Leaf Diseases" can be examined.
7-"The disease indicates impairment to normal plant state which enhances or interrupts its crucial operations like transpiration, photosynthesis, fertilization, pollination, and germination." with the sentence "The diseases tend to disablement for the classical state of plants which obstructs its imperative roles like photosynthesis, transpiration, pollination and so on." The sentences are very close in meaning. Sentence repetitions should be avoided.
8-Unecessary abbreviations should be avoided. "The majority of classification techniques include back propagation neural network (BPNN), support vector machine (SVM), decision tree (DT), artificial neural network (ANN), radial basis function (RBF), probabilistic neural network (PNN), are utilization for classifying the disease present in plant leaf"
Since RBF, PNN, and BPNN are not used in the rest of the article, these abbreviations should be removed.
9-The letter "t" is missing in the expression "Figure 2 provides LeNe". Correct it.

Experimental design

10-There are other Gaussian-based studies in the literature to eliminate or reduce noise in plant leaf images. "NCA-based hybrid convolutional neural network model for classification of cervical cancer on gauss-enhanced pap-smear images" and "Classification of OME with Eardrum Otoendoscopic Images Using Hybrid-Based Deep Models, NCA, and Gaussian Method"
In the study, the Gaussian technique used to reduce noise on the image should be mentioned. In addition, it should be explained why the "Adaptive Anisotropic diffusion" technique is used to eliminate noise.
11-It seems that the authors do not provide information about the limits of the CFSA algorithm. In this study, comprehensive information should be given about the constraints of the algorithm and the conditions under which the experiments were carried out. Number of iterations, number of candidate solutions, etc.
Additionally, the limitations of this study should be clearly stated.
12-The main purpose of using position and color augmentation techniques is not clearly discussed in the article. Why are these techniques needed?

Validity of the findings

13-Explain why f-measure (f-score), one of the most commonly used metrics, was not used in the experimental results. If there is no valid reason, please add it to table 1 and table 2.
14-The conclusion of the article should be improved.

---

## Round 0.2 · Minor Revisions

Dear author,

The reviews for your revised manuscript have been received. Your paper still needs minor revision. You will be expected to revise the paper according to the experimental design comments provided by reviewer 2.

Best wishes,

**Language Note:** The review process has identified that the English language must be improved. PeerJ can provide language editing services - please contact us at copyediting@peerj.com for pricing (be sure to provide your manuscript number and title). Alternatively, you should make your own arrangements to improve the language quality and provide details in your response letter. – PeerJ Staff

·

Basic reporting

no comment

Experimental design

no comment

Validity of the findings

no comment

Additional comments

no comment

Reviewer 2 ·

Basic reporting

The authors fulfilled all requests in the revision and reflected them in the article. However, a few things have been overlooked and need to be completed.
1- The spelling rules of the language in the text must be examined in detail.
2- It is not visible from which journal the sources added to the article in revision are cited. Additionally, volume and issue numbers should be checked and any errors should be corrected.

Experimental design

.

Validity of the findings

.

Additional comments

.

---

## Round 0.3 · accepted · Accept

Dear authors,

Thank you for the revision and for clearly addressing all the reviewers' comments. I confirm that the paper is improved and addresses the concerns of the reviewers. Your paper is now acceptable for publication in light of the last revision.

Best wishes,